# Development and Characterization of Fluorescent Protein-Tagged *Vibrio harveyi* Strains as a Versatile Tool for Studying Infection Dynamics and Strain Interactions

**DOI:** 10.3390/pathogens14030247

**Published:** 2025-03-03

**Authors:** Charalampos Chalmoukis, Stavros Droubogiannis, Vassiliki A. Michalopoulou, Adriana Triga, Panagiotis F. Sarris, Pantelis Katharios

**Affiliations:** 1Department of Biology, University of Crete, 71409 Heraklion, Crete, Greece; bchalmoukis@gmail.com (C.C.); stavros.drou@gmail.com (S.D.); triga@hcmr.gr (A.T.); p.sarris@imbb.forth.gr (P.F.S.); 2Institute of Marine Biology, Biotechnology and Aquaculture, Hellenic Centre for Marine Research, Gournes, 71500 Heraklion, Crete, Greece; 3Institute of Molecular Biology and Biotechnology, Foundation for Research and Technology-Hellas, 70013 Heraklion, Crete, Greece; v.michalopoulou@imbb.forth.gr; 4Department of Biosciences, College of Life and Environmental Sciences, University of Exeter, Exeter EX4 4PY, UK

**Keywords:** *Vibrio harveyi*, transconjugants, GFP, RFP, triparental mating

## Abstract

Fluorescent protein-tagged bacterial strains are widely used tools for studying host-pathogen interactions and microbial dynamics. In this study, we developed and characterized *Vibrio harveyi* strains genetically modified to express green fluorescent protein (GFP) and red fluorescent protein (RFP). These strains were constructed using triparental mating and evaluated for phenotypic, genomic, and virulence attributes. Genomic analyses revealed strain-specific variations, including mutations in key regulatory and metabolic genes, such as luxO and transketolase. While plasmid acquisition imposed metabolic costs, resulting in altered growth and antibiotic sensitivities in certain transconjugants, others demonstrated robust phenotypic stability. Virulence assays using gilthead seabream larvae revealed that most tagged strains retained moderate pathogenicity, with visualization of co-infections highlighting the potential for studying strain-specific interactions. Furthermore, fluorescent microscopy confirmed the successful colonization and localization of tagged bacteria within host tissues. These findings underscore the utility of GFP- and RFP-tagged *Vibrio harveyi* as versatile tools for infection dynamics, offering a foundation for future research on strain interactions and pathogen-host relationships.

## 1. Introduction

*Vibrio harveyi* is a significant pathogen in aquaculture, responsible for vibriosis—one of the most severe diseases affecting farmed fish and invertebrates. Aquatic diseases collectively cause economic losses exceeding $9 billion annually, with vibriosis alone contributing approximately $1 billion to these losses [1,2,3]. In fish, *V. harveyi* infections manifest with clinical signs such as skin erosion, epidermal ulceration, hemorrhages, panophthalmitis, and severe enteritis, resulting in considerable mortality and reduced productivity [2].

As a species, *V. harveyi* exhibits substantial genetic and phenotypic diversity, including variations in virulence. This diversity complicates disease management and enhances the bacterium’s adaptability to different environments and hosts [4,5]. Furthermore, *V. harveyi* is frequently isolated alongside other bacterial species in co-infections, emphasizing its role in complex disease dynamics [2,5]. Even in cases where *V. harveyi* is the sole pathogen, multiple distinct strains can often be recovered from a single diseased host, underscoring its genetic heterogeneity and ecological versatility.

*Vibrio harveyi* possesses a diverse arsenal of virulence factors that contribute to its pathogenicity. These include transporters, secretion systems, bacterial toxins, motility-associated proteins, exosome-associated proteins, and siderophores, which collectively facilitate host colonization, nutrient acquisition, and immune evasion [5]. Additionally, quorum sensing plays a central role in regulating virulence in *V. harveyi*, orchestrating the expression of these factors in response to cell density and environmental cues. This regulatory network emphasizes the adaptability and complexity of *V. harveyi* as a pathogen [6].

Fluorescent proteins (FPs) have become indispensable tools in live cell imaging, enabling the visualization of cellular processes in real time. These proteins fluoresce when excited by specific wavelengths of light, making them ideal for tracking protein interactions, nucleic acid dynamics, biochemical changes, and diverse cellular states [7,8,9]. Among the most widely used FPs are Green Fluorescent Protein (GFP), derived from the jellyfish *Aequorea victoria* [10], and Red Fluorescent Protein (RFP), originally isolated from corals of the genus *Discosoma* [11]. While early versions of RFP had limitations, such as slow chromophore maturation, advanced mutants like DsRed.T3 (DNT) now offer rapid maturation and strong fluorescence, making them highly suitable for experimental applications [12,13].

The use of FPs in bacterial pathogens has opened new avenues for studying their life cycles and interactions with hosts or other microbes [14,15]. We have applied molecular tagging with GFP and RFP to each of two different strains of *V. harveyi,* generating a total of four strains (each original strain tagged separately with GFP and RFP) to create a versatile model system. This model enables simultaneous tracking of multiple strains in mixed populations, allowing us to validate the tagged strains’ utility in future studies of microbial competition, antagonism, cooperation, or niche specialization within the same species. While this work focuses on the development and validation of the tagged strains, their potential applications highlight the broader significance of this tool for exploring strain-level dynamics in bacterial pathogens.

## 2. Materials and Methods

### 2.1. Bacterial Strains and Culture Conditions

The bacterial strains used for transconjugation were *Vibrio harveyi* Vhp1-sp and *Vibrio harveyi* VH2 [5]. The strains of *V. harveyi* are resistant to ampicillin 100 μg mL^−1^. The plasmid-carrying strains for transconjugation were *Escherichia coli* DH5α λpir pVSV102 (containing the GFP gene) and *Escherichia coli* DH5α λpir pVSV208 (containing the RFP gene) [13]. For the triparental mating method, *Escherichia coli* pRK2013 (kanamycin antibiotic resistance) was used as the helper strain. For the virulence test, *Phaeobacter piscinae* S26 was used as a negative control. According to the literature, it is a non-pathogenic bacterium with probiotic properties beneficial for the larvae [16].

*Escherichia coli* strains were cultured overnight (O/N) in Luria-Bertani (LB) medium at 37 °C. *Vibrio harveyi* strains were grown in various media depending on the experiment, including LB, LB* (10 g L^−1^ tryptone, 5 g/L yeast extract, 10 g L^−1^ NaCl, 1 L deionized water, 0.75 g L^−1^ MgSO_4_, 1.5 g L^−1^ KCl, 0.73 g L^−1^ CaCl_2_) and LB-2% NaCl. These strains grew effectively in O/N cultures at temperatures of 25, 28, and 37 °C, or room temperature (RT), as required by the experimental conditions. Bacterial strains were stored and maintained in MicroBank tubes at −80 °C.

### 2.2. Plasmid Transfer via Transcojugation

The plasmids pVSV102, encoding GFP and kanamycin resistance, and pVSV208, encoding DsRed-Express (RFP) and chloramphenicol resistance, were transferred from *Escherichia coli* to *Vibrio harveyi* strains (Vhp1-sp and VH2) via triparental mating, utilizing the conjugative *E. coli* strain pRK2013 as a helper, following the method described by Christiansen et al. (1997) [17].

Overnight cultures of donor, recipient, and helper cells were grown in LB medium with appropriate antibiotics (kanamycin, 50 μg mL^−1^ for pVSV102; chloramphenicol, 25 μg mL^−1^ for pVSV208) for *E. coli* strains, and in LB-2% NaCl for *V. harveyi* strains. These cultures were diluted 1:50 for *E. coli* and 1:100 for *V. harveyi* into fresh medium and incubated at 37 °C for 3 h to an OD_600_ of approximately 1.

To initiate mating, 250 μL of helper culture, 250 μL of donor culture, and 1 mL of recipient culture (ratio 1:1:4) were combined in a microcentrifuge tube. Control samples containing only donor, helper, or recipient cells were prepared similarly. The mixtures were centrifuged at 13,500 rpm for 3 min, supernatants were discarded, and pellets were resuspended in the residual liquid (approximately 10 μL). Resuspended mixtures were spotted onto LBS agar plates and incubated at 37 °C overnight.

Colonies from the mating spots were subsequently streaked onto LB-2% agar plates containing the selective antibiotics: kanamycin (50 μg mL^−1^) and ampicillin (100 μg mL^−1^) for pVSV102, or chloramphenicol (25 μg mL^−1^) and ampicillin (100 μg mL^−1^) for pVSV208. Plates were incubated overnight at 37 °C, and if no colonies were observed, the plates were further incubated at room temperature (~25 °C) overnight. Transconjugant colonies were picked the next day and restreaked on fresh selective plates to confirm purity and eliminate contamination. Control samples did not exhibit growth on selective media containing combined antibiotics, confirming the specificity of the method. Moreover, to assess plasmid stability, each strain underwent continuous subculturing by selecting single colonies and reculturing them over five successive passages. The plasmid remained stable in all transconjugants throughout these recultures.

### 2.3. Bacterial Growth Assessment

To evaluate potential growth differences between transconjugants and wild-type strains, bacterial growth was monitored over 22 h using a microplate reader Infinite PRO 200 (Tecan Trading AG, Männedorf, Switzerland) at 25 °C. Absorbance (OD_600_) measurements were recorded every 10 min. Overnight cultures were prepared in 5 mL LB* medium supplemented with appropriate antibiotics for transconjugants: kanamycin (50 μg mL^−1^) and ampicillin (100 μg mL^−1^) for pVSV102, and chloramphenicol (25 μg mL^−1^) and ampicillin (100 μg mL^−1^) for pVSV208. Cultures were diluted in LB* to a concentration of 2 × 10⁶ CFU mL^−1^ prior to use. In a 96-well microplate, 200 μL of LB* was added to the peripheral wells to minimize edge effects. For each strain, triplicate wells received 100 μL of LB* and 100 μL of the 2 × 10⁶ CFU mL^−1^ dilution, yielding a final concentration of 10⁶ CFU mL^−1^ per well. The plate was incubated, and absorbance readings were collected over the 22-h period. Each assay was conducted as two independent experiments, each with three technical replicates.

Initial bacterial concentrations of all strains were validated by colony counting following serial dilutions (10⁻⁴, 10⁻⁵, and 10⁻⁶) and plating. Colonies were counted after 12–16 h of incubation at 25 °C.

### 2.4. Motility Evaluation

The Motility Indole Ornithine (MIO) medium was utilized to evaluate bacterial motility. Isolated colonies from a pure overnight culture were collected and inoculated by stabbing the center of the medium to approximately half its depth. The tubes were then incubated for 24 h at 25 °C. Positive motility was indicated by the presence of turbidity or diffuse growth spreading from the stab line. Additionally, motility was verified by examining the samples under a Nikon Eclipse 50i microscope.

The swimming motility of the bacterial strains was assessed as described by Melanie Pearson [18] with minor modifications. Briefly, O/N bacterial cultures were diluted 1:100 into fresh LB* medium and incubated at 25 °C with shaking at 180 rpm until reaching a concentration of approximately 10^7^ CFU mL^−1^. A 5 µL aliquot of each bacterial culture was then spotted onto the center of 0.3% LB* agar plates. The plates were incubated at 25 °C for 24 h, and the diameter of the swimming zone was measured to evaluate motility. All assays were performed in triplicate to ensure the reproducibility of results.

### 2.5. Metabolic Profile

The GEN III Microplate™ (BIOLOG, Hayward, Berkeley Heights, NY, USA) was used to assess metabolic profile changes in the transconjugants compared to the wild-type (WT) strains, offering a comprehensive metabolic fingerprint based on their substrate utilization capabilities.

### 2.6. Antibiotic Susceptibility

Alterations in the antibiotic susceptibility profile of the transconjugants were evaluated using the antibiotic diffusion disc assay. The antibiotics selected for this study, commonly used in aquaculture, included oxalinic acid (OA, 2 μg), oxytetracycline (OT, 30 μg), florfenicol (FFC, 30 μg), flumequine (UB, 30 μg), and sulfamethoxazole/trimethoprim (SXT, 25 μg). Antibiotic susceptibility was tested on Mueller-Hinton agar supplemented with 2% NaCl (MH-2%) at 25 °C, with zones of inhibition measured after 48 h.

### 2.7. In Vivo Virulence Assessment

Potential changes in the virulence of the transconjugants resulting from the modification method applied were assessed using an infection challenge assay previously developed for gilthead seabream (*Sparus aurata*) larvae [19]. The eggs used for the assay were sourced from the broodstock facility of the Hellenic Center for Marine Research (HCMR). Eggs were observed under a stereoscope (Nikon SM800, Nikon Corporation, Tokyo, Japan) to ensure they were at the same developmental stage. Prior to the experiment, the eggs were washed three times with filtered and sterilized borehole seawater to ensure cleanliness and then transferred individually into the wells of sterile 96-well plates.

The infection challenge began immediately after hatching, corresponding to 0 days post-hatching (0 dph). Two negative controls were included one without any bacterial inoculum and another inoculated with *Phaeobacter piscinae* S26, a probiotic strain known for its beneficial effects. Overnight cultures of *Vibrio harveyi* strains and *Phaeobacter piscinae* S26 were prepared for the assay by centrifuging to separate the bacterial cells from the growth medium, followed by resuspension in sterile saline to achieve a final concentration of 10^6^ CFU mL^−1^. Larval mortality was monitored daily over a 5-day period.

### 2.8. Genomic DNA Extraction and Sequencing

DNeasy Blood and Tissue Kit (QIAGEN, Hilden, Germany) was utilized to extract the DNA of the bacterial strains according to the manufacturer’s protocol. The quality and quantity of the extracted DNA were assessed using a NanoDrop Spectrophotometer (ThermoFisher Scientific, Waltham, MA, USA) to measure purity (A260/A280 ratio) and concentration. Additionally, DNA integrity was evaluated via gel electrophoresis.

The bacterial genomes of the transconjugants were sequenced using a short-read approach. DNBseq technology was used. The library preparation process included several critical steps: DNA was fragmented and subjected to size selection, followed by end-repair and the addition of “A” tails. Adapters were then ligated to the fragments, and a second round of size selection was performed. This was followed by PCR amplification to enrich the library, and comprehensive quality control (QC) was conducted to ensure the library’s integrity. Low-quality reads and adapter-contaminated sequences were filtered from the raw data to ensure high-quality output. Data processing, including contamination removal and validation of usable reads, was carried out using SOAPnuke, a bioinformatics tool developed by BGI [20]. Reads have been deposited in the Genbank, see Data Availability section.

### 2.9. Genomic Variation Analysis

For the variation analysis, the raw sequence data of each of the transconjugants were aligned to the corresponding genome of wild-type *V. harveyi* VH2 or *V. harveyi* Vhp1-sp using BWA-mem aligner (http://bio-bwa.sourceforge.net/) (accessed on 15 December 2024) with strict parameters, followed by variant calling using FreeBayes [21], a Bayesian genetic variant detector. Unmapped reads were de novo assembled using Geneious assembler (Geneious Prime^®^ 2024.0.3) and confirmed to match the plasmid of interest.

### 2.10. In Vivo Fluorescence Imaging of Infection

To assess the suitability of the transconjugant bacterial strains for infectivity studies, gilthead seabream larvae were infected with RFP- and GFP-tagged bacteria, either individually or in combination. Infections were conducted as previously described, using fertilized gilthead seabream eggs placed in 96-well plates. However, larvae were infected either at hatching or at 3- or 4 days post-hatching. Moribund and deceased fish larvae were collected at different time points post-infection and examined under a fluorescent microscope (Olympus IX70, Olympus Inc., Tokyo, Japan) equipped with GFP (excitation: 470–490 nm, emission: 510–530 nm) and RFP (excitation: 540–560 nm, emission: 580–620 nm) filter sets. Micrographs were captured using a Calser camera with an exposure of 50–70 ms and processed in ImageJ 1.54g (National Institutes of Health, https://imagej.net/ij/) (accessed on 2 January 2025) to create composite images, overlaying fluorescent signals onto non-fluorescent backgrounds.

### 2.11. Statistical Analysis

Growth curves of the WT strains and their corresponding transconjugants were plotted over time. The curves were statistically compared using the Area Under the Curve (AUC) calculated using the Growthcurver package (v.0.3.1) in R. Statistical comparisons of AUCs were performed using one-way ANOVA in GraphPad Prism 10. In vivo, virulence was assessed using the Log-rank (Mantel-Cox) test, also conducted in GraphPad Prism 10 (GraphPad Software, Boston, MA, USA).

## 3. Results

### 3.1. Plasmid Transconjugation

The transconjugation with the plasmids of interest was successfully achieved for both *Vibrio harveyi* strains, VH2 and Vhp1-sp. For the transconjugation of strain Vhp1-sp, after initial incubation in the selective medium at 37 °C overnight (O/N), an additional incubation at room temperature overnight was required to obtain the genetically modified colonies. The transconjugant colonies were distinguishable by their coloration. Transconjugants carrying pVSV102 exhibited yellowish colonies, while those harboring pVSV208 displayed reddish colonies. These differed from the white colonies typical of *E. coli* strain [13] and the grayish-white colonies characteristic of *V. harveyi* strains [22]. The resulting transconjugant strains were as follows: *V. harveyi* VH2 pVSV102 (VH2 GFP), *V. harveyi* VH2 pVSV208 (VH2 RFP), *V. harveyi* Vhp1-sp pVSV102 (Vhp1-sp GFP) and *V. harveyi* Vhp1-sp pVSV208 (Vhp1-sp RFP). The transconjugants emit fluorescence under UV light, a characteristic that can be observed macroscopically on agar plates containing the modified bacteria, as shown in Figure 1.

### 3.2. Genomic and Variation Analysis in Fluorescent Protein—Tagged Strains

The analysis revealed that *V. harveyi* VH2, which has a 5,836,028 bp plasmid-free genome composed of two chromosomes, showed a single nucleotide substitution (G > A) in the gene coding for *LuxO* when carrying the plasmids pVSV102 and a single nucleotide substitution (C > T) in the same gene, when carrying the plasmid pVSV208. No additional variants were detected in both VH2 transconjugants.

In contrast, *V. harveyi* Vhp1-sp, whose genome consists of two chromosomes and two plasmids, exhibited more extensive variation. In the Vhp1-sp RFP strain, a total of 13 variants were identified, two of which were located in intergenic regions. Among the coding regions, two single nucleotide substitutions were identified in the gene encoding Mg-dependent DNAase, and a single substitution (T > C) was observed in the gene encoding transketolase. Similarly, in the Vhp1-sp GFP strain, 9 variants were detected, with one occurring in the intergenic region. This strain also shared the nucleotide substitutions in the Mg-dependent DNAase gene observed in the RFP strain. Additionally, a substitution (G > A) was detected in the gene encoding the T1SS peptidoglycan-associated lipoprotein LapL. Both Vhp1-sp transconjugants lost their native second plasmid pVh_VhP1-sp-2.

The identified variants exhibited high coverage and quality, underscoring their reliability and prevalence within the mutant populations (Appendix A).

### 3.3. Bacterial Growth Assessment

The genetic modification had no significant impact on the growth of Vhp1-sp, as the modified bacteria exhibited growth rates comparable to the wild type (WT). In contrast, both transconjugants of VH2 demonstrated reduced growth compared to the WT, with the reduction observed in the RFP strain being statistically significant. This is illustrated in the growth curves of the bacterial strains presented in Figure 2.

### 3.4. Motility Evaluation

All bacterial strains tested were motile in the MIO medium. In the swimming motility assay conducted on 0.3% LB agar, no statistically significant differences in swimming zone diameters were observed between transconjugants and their corresponding WT strains. The measured swimming zone diameters for the strains are summarized in Table 1. WT strains (Vhp1-sp and VH2) displayed swimming diameters of 25.00 ± 0.00 mm and 32.67 ± 3.79 mm, respectively. Fluorescent protein-tagged variants, including GFP and RFP-expressing derivatives, exhibited swimming diameters comparable to their corresponding wild-type strains. For example, Vhp1-sp GFP and RFP strains exhibited diameters of 26.67 ± 1.15 and 28.00 ± 2.00, while VH2 GFP and RFP strains showed diameters of 35.67 ± 9.50 and 43.33 ± 2.08 mm, respectively. These results suggest that the introduction of fluorescent protein markers does not significantly affect the motility of the bacterial strains.

### 3.5. Metabolic Profile

The modified bacteria exhibited changes in their metabolic profiles and chemical sensitivities compared to their corresponding WT strains. Specifically, VH2 GFP and VH2 RFP were unable to grow in the presence of lincomycin, tetrazolium violet, potassium tellurite, and sodium bromate. Additionally, VH2 GFP showed no growth in guanidine HCl, whereas Vhp1-sp GFP was capable of growth under the same conditions. Furthermore, Vhp1-sp RFP was unable to grow in sodium bromate, and VH2 RFP lost its resistance to aztreonam. The results are summarized in the Appendix A.

### 3.6. Antibiotic Susceptibility

The results of the antibiotic sensitivity assay are summarized in Table 2. While no major shifts in susceptibility were observed, some trends emerged among the transconjugants. In VH2, the RFP-labeled strain exhibited a tendency toward increased sensitivity to florfenicol, while the GFP-labeled strain remained comparable to the wild-type. For sulfamethoxazole/trimethoprim, minor variations were noted, but no clear pattern of increased or decreased sensitivity was evident. In Vhp1-sp, the transconjugants displayed largely similar antibiotic sensitivity profiles to the wild-type, with slight variations across different antibiotics. Notably, Vhp1-sp RFP showed a minor reduction in the inhibition zone for flumequine, though this change was not substantial. Overall, while some fluctuations were observed, they did not suggest a consistent trend in resistance or sensitivity shifts among the strains.

### 3.7. In Vivo Virulence Assessment

The group treated with the probiotic strain *Phaeobacter piscinae* S26 exhibited survival rates comparable to those of the untreated negative controls, with no statistically significant differences observed (Figure 3a). This suggests that the addition of live, non-pathogenic bacteria at the tested concentration does not adversely affect larval viability.

Comparisons between the groups treated with *Vibrio harveyi* WT strains and the saline-treated negative control revealed that both strains caused statistically significant increases in mortality compared to the saline control (Figure 3a). Further comparisons of the modified strains were made against both the saline-treated negative control and their respective wild-type (WT) strains (Figure 3b,c).

As summarized in Table 3, only the VH2 RFP strain demonstrated cumulative mortality levels comparable to its WT counterpart. All other modified strains exhibited cumulative mortality significantly lower than their respective WT strains. However, their mortality rates remained higher than those of the negative control, indicating that the modified strains retained virulence, causing mortality in a portion of the gilthead seabream larvae population.

### 3.8. In Vivo Fluorescent Imaging of Infection

The transconjugants of both *Vibrio harveyi* strains were readily detected and visualized in the colonized tissues of gilthead seabream larvae. In most cases, bacteria were observed within 1–2 days post-infection, predominantly localized in the eyes, on the skin near the tail, around the yolk sac, and, at later stages, in the developing intestine (Figure 4). Dead larvae appeared fully covered by bacteria (Figure 5).

## 4. Discussion

Despite extensive research efforts, the pathogenesis of *Vibrio harveyi* in various aquatic organisms remains poorly understood. This knowledge gap is largely attributed to the pathogen’s extraordinary adaptability, reflected in its significant phenotypic and genetic diversity. Fluorescent protein tagging has been widely applied to track and study bacteria during infection, providing insight into bacteria-host interactions. It has been successfully utilized in several members of the *Vibrio* genus, including human pathogens such as *Vibrio cholerae*, *V. parahaemolyticus*, and *V. vulnificus*, as well as aquatic pathogens like *V. harveyi*, *V. anguillarum* and *V. splendidus* [23,24]. Previous studies have demonstrated the effectiveness of GFP-tagged Vibrio strains in tracking infections, such as *V. harveyi* in abalone (*Haliotis tuberculata)* [25] and *Vibrio* species in Manila clam larvae [26]. Similarly, Wang et al. [27] used GFP tagging to investigate bacterial dissemination in Pacific oysters, highlighting its broad applicability in aquaculture research. Recent advances, including optimized fluorescent tagging methods [28] and successful tagging of *Vibrio splendidus* in sea cucumbers [29], further emphasize the versatility of this approach. The plasmids utilized in the current study were derived from the *Aliivibrio fischeri* plasmid pES213 [13]. Previous studies have demonstrated that both RFP- and GFP-labeled transconjugants in different *Vibrio* species exhibit high plasmid retention stability [13,23]. However, the retention of plasmids can impose a significant fitness cost on bacteria, potentially causing undesirable phenotypic changes that hinder their practical application. In some cases, modified strains can differ substantially from their wild-type counterparts, with notable differences in growth potential, antibiotic sensitivity, motility, and virulence. Aboubaker et al. [30] demonstrated that while GFP tagging of *Vibrio aestuarianus* enabled real-time infection tracking in Pacific oysters, it also led to significant phenotypic alterations in some strains. Specifically, GFP expression caused a weakly pathogenic strain to exhibit increased virulence, highlighting the potential metabolic or regulatory effects of plasmid-borne fluorescent markers. This is particularly critical when molecular tagging with fluorescent proteins is employed to trace pathogenic bacteria during infection and investigate their mechanisms of pathogenicity.

In this study, we have successfully modified two different strains of *Vibrio harveyi* by tagging each one of them with GFP and RFP using triparental mating. *Vibrio harveyi* appears to be more active and competent at higher temperatures [31], as indirectly demonstrated by the triparental mating method used in this experiment. Specifically, the results indicate that 37 °C is the optimal temperature, yielding faster and more promising outcomes for both strains. We hypothesize that the success at 37 °C can also be attributed to this temperature being optimal for *Escherichia coli* [32], which served as both the donor and helper strains in the triparental mating method. Since both *E. coli* and *V. harveyi* grow efficiently under these conditions, the plasmid transfer process was likely more effective.

The observed growth reduction in the VH2 strain transconjugants can be attributed to the fitness costs associated with plasmid carriage, as discussed by San Millan and MacLean [32]. Plasmids impose a metabolic burden on the host bacterium, which can manifest as reduced growth rates and competitiveness, especially in plasmid-free strains like VH2. The introduction of a new plasmid in VH2 likely required additional energy for plasmid retention and expression of GFP and RFP, leading to a statistically significant reduction in growth for the GFP transconjugant and a non-significant reduction for the RFP transconjugant. In contrast, the Vhp1-sp strain, which already harbors two plasmids, displayed no significant growth reduction in its transconjugants. This suggests that Vhp1-sp has evolved mechanisms to regulate its plasmid content and maintain an energy balance, possibly through compensatory evolution. The replacement of the smaller plasmid with the newly introduced one during transconjugation indicates a strategic regulation of plasmid content to minimize fitness costs. This aligns with the concept of compensatory evolution, where the initial fitness cost of plasmid carriage can be alleviated over time through mutations in the plasmid or host chromosome. Overall, the differences in growth effects between VH2 and Vhp1-sp transconjugants highlight the importance of pre-existing plasmid content and the host’s ability to adapt to new plasmids. A decrease in growth was also observed in *V. anguillarum* following GFP tagging via transposon mutagenesis. In this study, the authors similarly tagged a *V. splendidus* strain, which exhibited a slightly smaller reduction in growth. Both strains showed significant suppression of swimming motility, a phenomenon not observed in the strains analyzed in our study [27].

The phenotypic changes observed in the modified *Vibrio harveyi* strains using the BIOLOG GEN III assay could also be attributed to the metabolic and physiological consequences of plasmid acquisition. The inability of VH2 GFP and RFP transconjugants to grow in the presence of compounds like lincomycin, tetrazolium violet, potassium tellurite, and sodium bromate suggests a metabolic burden associated with plasmid retention and GFP/RFP expression. This burden likely disrupts critical pathways involved in stress tolerance and chemical resistance. For example, oxidative stress induced by GFP/RFP production could exacerbate sensitivity to oxidizing agents such as sodium bromate, while resource competition may impair protein synthesis, affecting antibiotic resistance mechanisms. These findings align with the well-documented phenomenon of fitness costs associated with plasmid carriage, particularly in plasmid-free strains like VH2, as discussed by San Millan and MacLean [33]. Interestingly, the Vhp1-sp transconjugants displayed fewer phenotypic changes and maintained robust growth under similar conditions, likely due to the strain’s pre-existing plasmid adaptation. The observed tolerance to guanidine HCl in Vhp1-sp GFP, compared to its absence in VH2 GFP, highlights the role of strain-specific metabolic capacities and compensatory evolution. Vhp1-sp appears to have evolved regulatory mechanisms to balance the metabolic demands of multiple plasmids, enabling it to accommodate new plasmid acquisitions with minimal disruption.

The antibiotic sensitivity assay revealed subtle variations in susceptibility between the two *Vibrio harveyi* strains following plasmid acquisition, highlighting the influence of host background and plasmid integration on antimicrobial responses. In the VH2 transconjugants, the RFP-tagged strain exhibited a tendency toward increased sensitivity to florfenicol, while the GFP-tagged strain remained comparable to the wild-type. This is consistent with the idea that plasmid acquisition can impair cellular functions, such as efflux pump activity or folate synthesis pathways, which are targeted by these antibiotics. In contrast, the Vhp1-sp transconjugants displayed largely similar antibiotic sensitivity profiles to the wild-type, with only a slight reduction in the flumequine inhibition zone in the RFP-tagged strain. The absence of major shifts in resistance suggests that Vhp1-sp, which already harbors multiple plasmids, may have regulatory mechanisms that minimize metabolic disruptions. Additionally, the new plasmid could interact synergistically with pre-existing resistance genes or influence gene expression, thereby conferring a selective advantage in specific conditions. These findings emphasize that while plasmid acquisition can introduce metabolic burdens and minor susceptibility changes, the extent of its impact varies depending on the host strain’s genetic background and adaptive potential.

The genomic analysis revealed significant differences in the genetic responses of *Vibrio harveyi* VH2 and Vhp1-sp to plasmid acquisition, providing insights into the underlying mechanisms driving their phenotypic changes. In VH2, the single nucleotide substitution (G to A) in the *luxO* gene suggests targeted alterations in quorum sensing regulation, a key system in *Vibrio* species that controls diverse functions, including metabolism and stress response [34]. This substitution may have also contributed to the observed phenotypic changes, such as altered antibiotic sensitivities and reduced growth under specific conditions, by disrupting the precise regulation of metabolic pathways. The absence of additional mutations in VH2 transconjugants underscores the plasmid’s direct impact on the strain’s physiology without extensive genomic restructuring. In contrast, the Vhp1-sp genome exhibited more extensive variation, likely due to its pre-existing plasmid content and ability to regulate multiple plasmids. The substitutions in the Mg-dependent DNAase gene and transketolase gene in both RFP and GFP strains may have affected DNA repair and central carbon metabolism, respectively, potentially enhancing the strain’s capacity to adapt to the plasmid’s metabolic burden. Transketolase catalyzes reactions of the nonoxidative branch of the pentose-phosphate pathway (PPP) [35]. The observed mutation in the gene coding transketolase may reflect a shift in the metabolic flux of the bacteria in order to accommodate the energetic demands of maintaining the plasmid. In addition, the observed mutation in the Mg-dependent DNase likely contributed to the dynamics of plasmid retention and loss. Mg-dependent DNases are known to play a critical role in degrading extracellular or foreign DNA, such as plasmids, thereby regulating DNA acquisition and maintenance [36,37]. A loss-of-function mutation in this enzyme could reduce the degradation of the newly acquired plasmid of interest, enhancing its stability and retention within the transconjugant.

Moreover, the mutation may have indirectly contributed to the loss of the smallest plasmid. It is known that plasmid incompatibility often leads to competition for replication and partitioning systems, and the altered DNase activity might disrupt this balance, favoring the stabilization of one plasmid over another [38]. Additionally, the reduced DNase activity could reflect an adaptive response to reduce cellular stress or metabolic burden associated with plasmid acquisition, further supporting the preferential retention of the plasmid of interest. These observations emphasize the complex interplay between DNA maintenance mechanisms and plasmid dynamics in bacterial systems.

Furthermore, the mutation in the *lapL* gene encoding a T1SS peptidoglycan-associated lipoprotein in Vhp1-sp GFP suggests potential impacts on cell envelope integrity and protein secretion, which could explain the differences in antibiotic resistance and stress tolerance observed between the GFP and RFP transconjugants.

The reduced virulence in most transconjugants suggests that plasmid acquisition and genetic changes impact pathogenicity. As already stated, plasmid carriage imposes a metabolic burden on bacteria, redirecting energy and resources away from virulence-associated functions to sustain plasmid replication and expression of GFP or RFP. In VH2 GFP and Vhp1-sp transconjugants, this trade-off likely contributed to their reduced ability to cause mortality in gilthead seabream larvae. Additionally, the single nucleotide substitution in *luxO* in VH2 GFP could have disrupted quorum sensing, a key regulator of virulence in Vibrio species, further attenuating the strain’s pathogenicity. In Vhp1-sp transconjugants, the more extensive genomic alterations, including mutations in genes such as Mg-dependent DNAase, transketolase, and *lapL*, may have compromised essential virulence traits such as DNA repair, central metabolism, and cell envelope integrity, respectively, thus reducing their ability to establish infections.

Interestingly, VH2 RFP retained its virulence, suggesting that the specific interaction between the plasmid and the host genome in this strain preserved its pathogenic potential. This could be due to the relatively minimal fitness cost observed in VH2 RFP compared to VH2 GFP, as indicated by the phenotypic data, or the possibility that *luxO* mutation had a lesser impact on virulence in the RFP strain. Despite the reduced virulence of most transconjugants, their ability to cause mortality above negative controls demonstrates that the genetic and phenotypic changes did not entirely abolish pathogenicity, emphasizing the robust nature of *Vibrio harveyi* as a pathogen. It is remarkable, that even in the gilthead seabream larval model, bacteria colonization was localized in the eye, the intestine and skin, particularly in the tail region, organs most affected during epizootics of *Vibrio harveyi* in larger fish [2]. Of particular significance, both GFP- and RFP-labeled strains of the same species could be observed in co-infections due to their distinct fluorescent signals, which could be utilized in future studies, such as investigations of strain-specific interactions or bacterial antagonism. Ultimately, this methodology could be extended to other marine pathogens such as *Vibrio parahaemolyticus, Vibrio anguillarum, and Photobacterium damselae* enabling comparative studies on infection mechanisms across species. Fluorescent-tagged bacteria provide the ability to track bacterial colonization in real-time which makes them an ideal tool for studying pathogen persistence, transmission routes, and interaction with host and environment.

## Figures and Tables

**Figure 1 pathogens-14-00247-f001:**
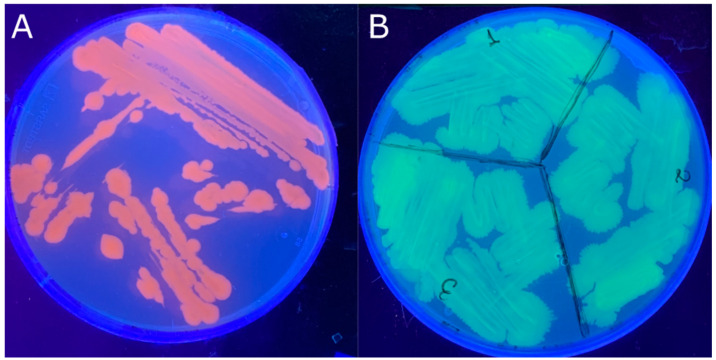
Plates with genetically modified *Vibrio harveyi* strains (**A**) Vhp1-spRFP (**B**) VH2 GFP under UV light.

**Figure 2 pathogens-14-00247-f002:**
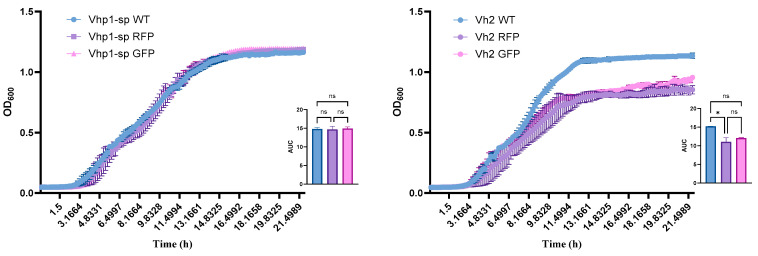
Bacterial growth over a 22-h period is shown for Vhp1-sp (**left**) and VH2 (**right**) alongside their respective RFP and GFP transconjugants. Data are presented as mean ± SE. The inset graphs display the area under the curve (AUC) for the growth curves, with statistical significance indicated at *p* < 0.05 (asterisk indicates statistical significance, ns: not significant).

**Figure 3 pathogens-14-00247-f003:**
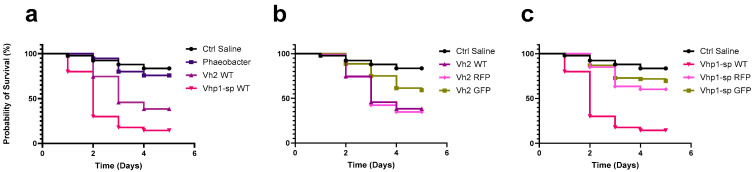
Survival curves of gilthead seabream larvae over a 5-day period. (**a**) Comparison of negative controls (saline and *Phaeobacter* S26) with larvae infected by pathogenic *Vibrio harveyi* wild-type strains. (**b**) Survival analysis of larvae exposed to saline control and *V. harveyi* strain VH2 wild-type (WT) along with its corresponding RFP- and GFP-labeled transconjugants. (**c**) Survival analysis of larvae exposed to saline control and *V. harveyi* strain Vhp1 wild-type (WT) along with its corresponding RFP- and GFP-labeled transconjugants.

**Figure 4 pathogens-14-00247-f004:**
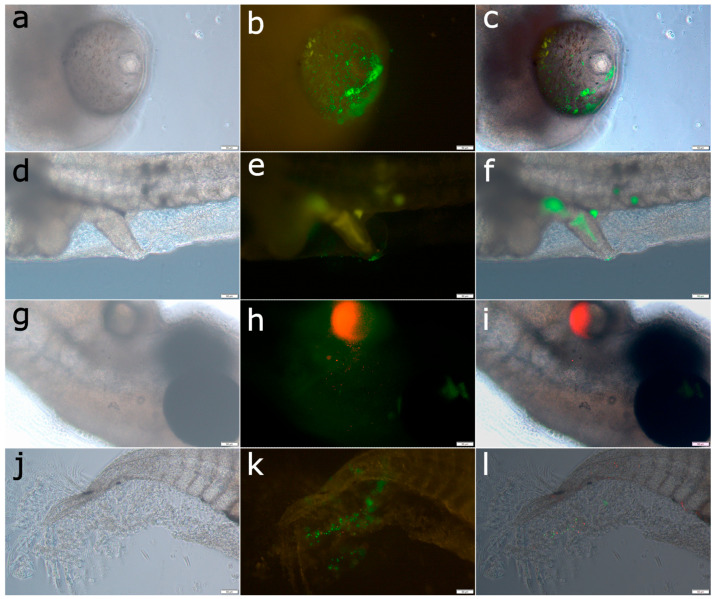
Visualization of fluorescent transconjugants of *Vibrio harveyi* strains during infection of gilthead seabream larvae. The left column shows images captured with conventional light microscopy, the middle column displays images taken with fluorescent microscopy, and the right column presents composite images. (**a**–**c**) Eye of a larva challenged at hatching with Vhp1-sp GFP, observed on day 3 post-infection. (**d**–**f**) Intestine of a larva challenged at hatching with VH2 GFP, observed on day 4 post-infection. (**g**–**i**) Colonization of the skin and yolk area by Vhp1-sp RFP bacteria in a larva challenged at 4 days post-hatch and observed 2 days later. (**j**–**l**) Co-infection of the larval tail by VH2 GFP and VH2 RFP bacteria, challenged at hatching and observed on day 3 post-infection. Scale bars in the bottom right corners of each panel represent a magnification of 50 μm.

**Figure 5 pathogens-14-00247-f005:**
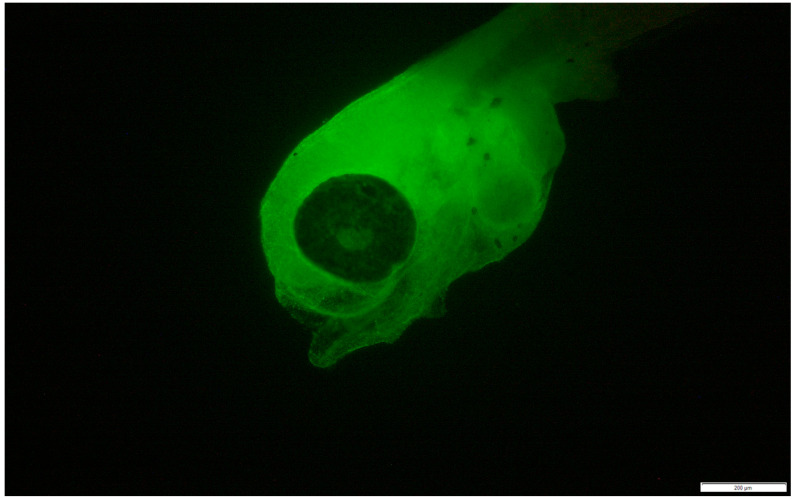
A deceased gilthead seabream larva covered by GFP-tagged *Vibrio harveyi* strain VH2. The larva was infected at 3 days post-hatch (dph) and died 2 days later. Scale bars in the bottom right corners of each panel represent a magnification of 200 μm.

**Table 1 pathogens-14-00247-t001:** Swimming zone diameter (mm) of *V. harveyi* strains in 0.3% LB agar plates.

Bacterial Strains	Swimming Zone Diameter (mm)
Vhp1-sp (WT)	25.00 ± 0.00
Vhp1-sp GFP	26.67 ± 1.15
Vhp1-sp RFP	28.00 ± 2.00
VH2 (WT)	32.67 ± 3.79
VH2 GFP	35.67 ± 9.50
VH2 RFP	43.33 ± 2.08

**Table 2 pathogens-14-00247-t002:** Zones of inhibition (mm) of *V. harveyi* strains in aquaculture-relevant antibiotics.

Strains	Zones of Inhibition (mm)				
	OA 2 μg	OT 30 μg	FFC 30 μg	UB 30 μg	SXT 25 μg
VH2 WT	16 ± 2.2	20 ± 0.5	24 ± 1.2	23 ± 1.9	18 ± 1
VH2 GFP	12 ± 2.1	22 ± 0.8	25 ± 0.9	21 ± 1	17 ± 1.2
VH2 RFP	15 ± 1.6	23 ± 0.8	29 ± 1.7	25 ± 0.8	20 ± 2
Vhp1-sp WT	16 ± 0.5	22	26 ± 0.5	24 ± 0.5	19 ± 0.5
Vhp1-sp GFP	16 ± 0.5	24 ± 2.4	27 ± 0.5	22 ± 0	18 ± 0.8
Vhp1-sp RFP	15 ± 2.1	22 ± 0.9	25 ± 0.5	24 ± 0.9	20 ± 0.8

**Table 3 pathogens-14-00247-t003:** The cumulative mortality rates of gilthead seabream larvae exposed to various bacterial strains tested.

Treatment	Cumulative Mortality (%)	Compared to Control	Compared to VH2 (WT)
Control	16.484	-	χ^2^ (185, 1) = 38.79, *p* < 0.0001
VH2 (WT)	61.702	χ^2^ (185, 1) = 38.79, *p* < 0.0001	-
VH2 GFP	84.374	χ^2^ (179, 1) = 11.98, *p* = 0.0005	χ^2^ (182, 1) = 10.36, *p* = 0.0013
VH2 RFP	65.217	χ^2^ (183, 1) = 43.98, *p* < 0.0001	χ^2^ (186, 1) = 0.1729, *p* = 0.6776
**Treatment**	**Cumulative Mortality (%)**	**Compared to Control**	**Compared to Vhp1-sp (WT)**
Control	16.484	-	χ^2^ (181, 1) = 97.17, *p* < 0.0001
Vhp1-sp (WT)	85.556	χ^2^ (181, 1) = 97.17, *p* < 0.0001	-
Vhp1-sp GFP	30.435	χ^2^ (183, 1) = 4.794, *p* = 0.0286	χ^2^ (182, 1) = 71.59, *p* < 0.0001
Vhp1-sp RFP	39.785	χ^2^ (184, 1) = 11.80, *p* = 0.0006	χ^2^ (183, 1) = 59.35, *p* < 0.0001

## Data Availability

Clean reads of the transconjugants produced from this project were submitted to the Sequence Read Archive (SRA) of the NCBI in Biosamples: SAMN46357331 for VH2 pVSV102 (VH2 GFP), SAMN46357332 for VH2 pVSV208 (VH2 RFP), SAMN46358779 for VhP1-sp pVSV102 (VhP1-sp GFP), SAMN46358780 for VhP1-sp pVSV208 (VhP1-sp RFP). The complete genomes of the wild-type strains have been deposited in the Genbank, with the accession numbers CP169261 and JAIWIW000000000, for VH2 and VhP1-sp, respectively.

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
