# Peer review of "Development and Characterization of Fluorescent Protein-Tagged Vibrio harveyi Strains as a Versatile Tool for Studying Infection Dynamics and Strain Interactions"

_pathogens, 2025, doi:10.3390/pathogens14030247_

Round 1

Reviewer 1 Report

Comments and Suggestions for Authors

This paper “Development and characterization of fluorescent protein-tagged Vibrio harveyi strains as a versatile tool for studying infection dynamics and strain interactions” is well-structured. The use of multiple bacterial strains, triparental mating, and genomic analysis enhances credibility. Secondly, the integration of fluorescence imaging and in vivo infection studies provides valuable insights into pathogen-host interactions.

However, there are some questions need to answer and some places for improvement:

  1. 1. The discussion on fitness cost due to plasmid retention is insightful, but it would be beneficial to compare this to other bacterial tagging studies. I suggested including more literatures on how other Vibrio species or bacterial pathogens handle fluorescence tagging. It will be better to give more introduction and discussion for this.
  2. Line 67-68: “we have applied molecular tagging with GFP and RFP to two different strains of Vibrio harveyi to create a versatile model system”

Please make the information more clear here, by mention which strain for GFP and RFP respectively.

  1. 3. Line 456-458: “The observed reduction in virulence of most transconjugants compared to their wild-type counterparts reflects the interplay between plasmid acquisition, phenotypic changes, and genetic alterations”

I suggested to change into:  "The reduced virulence in most transconjugants suggests that plasmid acquisition and genetic changes impact pathogenicity."

4.Line 162-163: Care was taken to ensure that all eggs were at the same developmental stage.

Author may need to clarify what kind of care specifically to ensure that all eggs were at the same developmental stage?

  1. Figure 2: In order to make the figure be distinguished more easily for the reader, I suggested to use green color for the GFP-tagged strain and red color for the RFP-tagged strain.
  2. 6. Line 239-243: Does any virulence factor related with all these variants in the RFP and GFP strains? If yes, how many?
  3. 7. Table 2: Is there any possibility to analyze this data statistically?

Author Response

This paper “Development and characterization of fluorescent protein-tagged Vibrio harveyi strains as a versatile tool for studying infection dynamics and strain interactions” is well-structured. The use of multiple bacterial strains, triparental mating, and genomic analysis enhances credibility. Secondly, the integration of fluorescence imaging and in vivo infection studies provides valuable insights into pathogen-host interactions.

However, there are some questions need to answer and some places for improvement:

  1. 1. The discussion on fitness cost due to plasmid retention is insightful, but it would be beneficial to compare this to other bacterial tagging studies. I suggested including more literatures on how other Vibrio species or bacterial pathogens handle fluorescence tagging. It will be better to give more introduction and discussion for this.

Answer: We thank the reviewer for his/her suggestion. We now included the recommendations in the discussion in lines 365-371. It is now stated that Previous studies have demonstrated the effectiveness of GFP-tagged Vibrio strains in tracking infections, such as V. harveyi in abalone (Haliotis tuberculata) [25] and Vibrio species in Manila clam larvae [26]. Similarly, Wang et al. [27] used GFP-tagging to investigate bacterial dissemination in Pacific oysters, highlighting its broad applicability in aquaculture research. Recent advances, including optimized fluorescent tagging methods [28] and successful tagging of Vibrio splendidus in sea cucumbers [29], further emphasize the versatility of this approach.” And in lines 379-380, “Aboubaker et al. [30] demonstrated that while GFP-tagging of Vibrio aestuarianus enabled real-time infection tracking in Pacific oysters, it also led to significant phenotypic alterations in some strains.”.

Line 67-68: “we have applied molecular tagging with GFP and RFP to two different strains of Vibrio harveyi to create a versatile model system”

Please make the information more clear here, by mention which strain for GFP and RFP respectively.

Answer: We thank the reviewer for noting this out. In order to make it clearer for the reader, it is now stated on lines 66-69 that “We have applied molecular tagging with GFP and RFP to each of two different strains of V. harveyi, generating a total of four strains (each original strain tagged separately with GFP and RFP) to create a versatile model system”.

  1. 3. Line 456-458: “The observed reduction in virulence of most transconjugants compared to their wild-type counterparts reflects the interplay between plasmid acquisition, phenotypic changes, and genetic alterations”

I suggested to change into:  "The reduced virulence in most transconjugants suggests that plasmid acquisition and genetic changes impact pathogenicity."

Answer: We thank the reviewer for his/her suggestion. The sentence was adjusted accordingly.

4.Line 162-163: Care was taken to ensure that all eggs were at the same developmental stage.

Author may need to clarify what kind of care specifically to ensure that all eggs were at the same developmental stage?

Answer: We thank the reviewer for pointing this out. In order to make it clearer we updated the manuscript accordingly. It is now stated in lines 166-168 “Eggs were observed under a stereoscope (Nikon SM800, Nikon Corporation, Tokyo, Japan) to ensure they were at the same developmental stage”.

Moreover, in HCMR we have our own gilthead broodstock and daily production of eggs. Usually within each batch there are two developmental stages but in most cases (if not always), 80% of the eggs belong to one stage and 20% to the other, therefore using one stage only is feasible if screening is done with a stereoscope.

  1. Figure 2: In order to make the figure be distinguished more easily for the reader, I suggested to use green color for the GFP-tagged strain and red color for the RFP-tagged strain.

Answer: We appreciate the reviewer’s suggestion regarding the color scheme for better distinction between GFP- and RFP-tagged strains. Our initial approach was indeed to use green for GFP-tagged strains and red for RFP-tagged strains. However, we decided against this choice to accommodate readers with color vision deficiencies (color blindness where green and red is not distinguishable). To ensure accessibility and clarity for a wider audience, we opted for alternative colors that provide better contrast and differentiation for all readers.

  1. 6. Line 239-243: Does any virulence factor related with all these variants in the RFP and GFP strains? If yes, how many?

Answer: We thank the reviewer for his/her suggestion. This has already been discussed on lines 450–473, where we describe the mutations identified in the transconjugants and their potential impact on virulence. Specifically, we highlight that the luxO gene mutation in VH2 transconjugants is linked to quorum sensing regulation, a key system controlling virulence in Vibrio species. Additionally, for Vhp1-sp transconjugants, mutations in the Mg-dependent DNAase and lapL genes are mentioned as potential contributors to virulence-related functions. This discussion provides an overview of how genetic variations in the RFP and GFP strains may influence virulence.

  1. 7. Table 2: Is there any possibility to analyze this data statistically?

Answer: We thank the reviewer for noting this out. We have repeated this assay using replicates and analyzed this data statistically and are now represented properly in Table 2. The corresponding result paragraph (lines 294-304) was changed appropriately and is now stated “The results of the antibiotic sensitivity assay are summarized in Table 2. While no major shifts in susceptibility were observed, some trends emerged among the transconjugants. In VH2, the RFP-labeled strain exhibited a tendency toward increased sensitivity to florfenicol, while the GFP-labeled strain remained comparable to the wild-type. For sulfamethoxazole/trimethoprim, minor variations were noted, but no clear pattern of increased or decreased sensitivity was evident. In Vhp1-sp, the transconjugants displayed largely similar antibiotic sensitivity profiles to the wild-type, with slight variations across different antibiotics. Notably, Vhp1-sp RFP showed a minor reduction in inhibition zone for flumequine, though this change was not substantial. Overall, while some fluctuations were observed, they did not suggest a consistent trend in resistance or sensitivity shifts among the strains.” Moreover, the discussion was changed accordingly in lines 434-449 to “The antibiotic sensitivity assay revealed subtle variations in susceptibility between the two Vibrio harveyi strains following plasmid acquisition, highlighting the influence of host background and plasmid integration on antimicrobial responses. In the VH2 trans-conjugants, the RFP-tagged strain exhibited a tendency toward increased sensitivity to florfenicol, while the GFP-tagged strain remained comparable to the wild-type. This is consistent with the idea that plasmid acquisition can impair cellular functions, such as efflux pump activity or folate synthesis pathways, which are targeted by these antibiotics. In contrast, the Vhp1-sp transconjugants displayed largely similar antibiotic sensitivity pro-files to the wild-type, with only a slight reduction in flumequine inhibition zone in the RFP-tagged strain. The absence of major shifts in resistance suggests that Vhp1-sp, which already harbors multiple plasmids, may have regulatory mechanisms that minimize metabolic disruptions. Additionally, the new plasmid could interact synergistically with pre-existing resistance genes or influence gene expression, thereby conferring a selective advantage in specific conditions. These findings emphasize that while plasmid acquisition can introduce metabolic burdens and minor susceptibility changes, the extent of its impact varies depending on the host strain’s genetic background and adaptive potential.”.

Reviewer 2 Report

Comments and Suggestions for Authors

This study presents the development and characterization of genetically modified Vibrio harveyi strains expressing green fluorescent protein (GFP) and red fluorescent protein (RFP). The modifications were achieved via triparental mating, and their phenotypic, genomic, and virulence attributes were assessed. The research provides useful insights into the pathogenicity and strain interactions of V. harveyi, which is relevant for aquaculture and marine biology. The manusscript could benefit from the following ssuggessstionss: 

1: Introduction

- Provide a more comprehensive discussion of previous fluorescent tagging methods in Vibrio species. Also, mention the limitations of previous tracking methods and how this study addresses them.

2-Methods

-Clarify if plasmid stability was tested across multiple bacterial passages. Also, expand on fluorescence microscopy settings and imaging conditions to improve reproducibility.

3-Results

Further discuss how genomic variations might influence virulence and antibiotic susceptibility. Also, provide additional explanation for metabolic profiling differences between strains.

4-Discussion

Expand on how this tool could be used in future aquaculture research. Discuss the potential limitations of fluorescence-tagged strains, including any fitness costs.

5-Conclusion

-Provide recommendations for further research on the use of tagged Vibrio harveyi strains in disease control strategies. Also, discuss how this model could be applied to other pathogenic bacteria in marine environments.

Comments on the Quality of English Language

The manuscript is well-structured and professionally written, but minor grammatical issues, tense inconsistencies, and redundant phrasing should be addressed to enhance clarity and readability.

Author Response

This study presents the development and characterization of genetically modified Vibrio harveyi strains expressing green fluorescent protein (GFP) and red fluorescent protein (RFP). The modifications were achieved via triparental mating, and their phenotypic, genomic, and virulence attributes were assessed. The research provides useful insights into the pathogenicity and strain interactions of V. harveyi, which is relevant for aquaculture and marine biology. The manuscript could benefit from the following suggestions: 

1: Introduction

- Provide a more comprehensive discussion of previous fluorescent tagging methods in Vibrio species. Also, mention the limitations of previous tracking methods and how this study addresses them.

Answer: We thank the reviewer for his/her suggestion. We now included the recommendations in the discussion in lines 365-371. It is now stated that Previous studies have demonstrated the effectiveness of GFP-tagged Vibrio strains in tracking infections, such as V. harveyi in abalone (Haliotis tuberculata) [25] and Vibrio species in Manila clam larvae [26]. Similarly, Wang et al. [27] used GFP-tagging to investigate bacterial dissemination in Pacific oysters, highlighting its broad applicability in aquaculture research. Recent advances, including optimized fluorescent tagging methods [28] and successful tagging of Vibrio splendidus in sea cucumbers [29], further emphasize the versatility of this approach.” And in lines 379-380, “Aboubaker et al. [30] demonstrated that while GFP-tagging of Vibrio aestuarianus enabled real-time infection tracking in Pacific oysters, it also led to significant phenotypic alterations in some strains.”.

2-Methods

-Clarify if plasmid stability was tested across multiple bacterial passages. Also, expand on fluorescence microscopy settings and imaging conditions to improve reproducibility.

Answer: We thank the reviewer for pointing this out. The plasmid stability was indeed tested across multiple bacterial passages. It is now stated in the manuscript on lines 116-119. “Moreover, to assess plasmid stability, each strain underwent continuous subculturing by selecting single colonies and reculturing them over five successive passages. The plasmid remained stable in all transconjugants throughout these recultures.” Regarding the fluorescence microscopy settings and imaging conditions, in the manuscript is now stated on lines 208-213 that “Moribund and deceased fish larvae were collected at different time points post-infection and examined under a fluorescent microscope (Olympus IX70, Olympus Inc, Tokyo, Ja-pan) equipped with GFP (excitation: 470–490 nm, emission: 510–530 nm) and RFP (excita-tion: 540–560 nm, emission: 580–620 nm) filter sets.  Micrographs were captured using a Calser camera with an exposure of 50-70 ms and processed in ImageJ (National Institutes of Health, https://imagej.net/ij/) to create composite images, overlaying fluorescent signals onto non-fluorescent backgrounds.”

3-Results

Further discuss how genomic variations might influence virulence and antibiotic susceptibility. Also, provide additional explanation for metabolic profiling differences between strains.

Answer: We thank the reviewer for his/her suggestion. However, in the discussion it is stated "The genomic analysis revealed significant differences in the genetic responses of Vibrio harveyi VH2 and Vhp1-sp to plasmid acquisition, providing insights into the underlying mechanisms driving their phenotypic changes. In VH2, the single nucleotide substitution (G to A) in the luxO gene suggests targeted alterations in quorum sensing regulation, a key system in Vibriospecies that controls diverse functions, including metabolism and stress response [29].”

4-Discussion

Expand on how this tool could be used in future aquaculture research. Discuss the potential limitations of fluorescence-tagged strains, including any fitness costs.

Answer: We thank the reviewer for his/her suggestion. However, the limitations have already been discussed throughout the discussion. Specifically future applications of fluorescence-tagged strains are discussed in the last paragraph of the discussion, where we highlight their potential for studying host-pathogen interactions, strain-specific interactions, and bacterial antagonism. This can be found on lines 498–512. Fitness costs and their impact are extensively covered in multiple parts of the discussion. For example, the growth reduction in transconjugants is discussed on lines 395–415 while the metabolic burden of plasmid carriage is analyzed on lines 416–433. Moreover, the impact on antibiotic susceptibility and genetic variations is covered on lines 434-473 and the effect on virulence and potential compensatory evolution is discussed on lines 486–504.

5-Conclusion

-Provide recommendations for further research on the use of tagged Vibrio harveyi strains in disease control strategies. Also, discuss how this model could be applied to other pathogenic bacteria in marine environments.

Answer: We thank the reviewer for his/her suggestions. We now included in the manuscript in lines 510-516: “Ultimately, this methodology could be extended to other marine pathogens such as Vibrio parahaemolyticus, Vibrio anguillarum and Photobacterium damselae enabling comparative studies on infection mechanisms across species. Fluorescent-tagged bacteria provide the ability to track bacterial colonization real-time which makes them an ideal tool for studying pathogen persistence, transmission routes and interaction with host and environment.”.

Reviewer 3 Report

Comments and Suggestions for Authors

This study presents the development and characterization of Vibrio harveyi strains tagged with GFP and RFP to facilitate research on microbial dynamics and host-pathogen interactions. The authors employ triparental mating for strain construction and comprehensively analyze phenotypic, genomic, and virulence attributes. The work is well-structured, scientifically relevant, and contributes valuable tools for infection studies. The findings on metabolic costs, virulence retention, and strain-specific interactions offer a suitable foundation for further research.

Indeed, I have a minor comment.

Minor revision:

How was the extracted DNA validated? This should be included in the manuscript.

Author Response

This study presents the development and characterization of Vibrio harveyi strains tagged with GFP and RFP to facilitate research on microbial dynamics and host-pathogen interactions. The authors employ triparental mating for strain construction and comprehensively analyze phenotypic, genomic, and virulence attributes. The work is well-structured, scientifically relevant, and contributes valuable tools for infection studies. The findings on metabolic costs, virulence retention, and strain-specific interactions offer a suitable foundation for further research.

Indeed, I have a minor comment.

Minor revision:

How was the extracted DNA validated? This should be included in the manuscript.

Answer: We thank the reviewer for pointing this out. In the manuscript it is now stated in lines 179-184: “The quality and quantity of the extracted DNA were assessed using a NanoDrop Spectrophotometer (ThermoFisher Scientific, Waltham, Massachusetts, USA) to measure purity (A260/A280 ratio) and concentration. Additionally, DNA integrity was evaluated via gel electrophoresis."